# Context-Aware Saliency Guided Radiomics: Application to Prediction of Outcome and HPV-Status from Multi-Center PET/CT Images of Head and Neck Cancer

**DOI:** 10.3390/cancers14071674

**Published:** 2022-03-25

**Authors:** Wenbing Lv, Hui Xu, Xu Han, Hao Zhang, Jianhua Ma, Arman Rahmim, Lijun Lu

**Affiliations:** 1School of Biomedical Engineering, Southern Medical University, 1023 Shatai Road, Guangzhou 510515, China; bing179@smu.edu.cn (W.L.); xuhui989@smu.edu.cn (H.X.); hanxv8826@gmail.com (X.H.); jhma@smu.edu.cn (J.M.); 2Guangdong Provincial Key Laboratory of Medical Image Processing, Southern Medical University, 1023 Shatai Road, Guangzhou 510515, China; 3Guangdong Province Engineering Laboratory for Medical Imaging and Diagnostic Technology, Southern Medical University, 1023 Shatai Road, Guangzhou 510515, China; 4Pazhou Lab, Guangzhou 510330, China; 5Department of Medical Imaging, Nanfang Hospital, Southern Medical University, 1023 Shatai Road, Guangzhou 510515, China; zhossica@foxmail.com; 6Department of Integrative Oncology, BC Cancer Research Institute, 675 West 10th Avenue, Vancouver, BC V5Z 1L3, Canada; arahmim@bccrc.ca; 7Department of Radiology, University of British Columbia, 2211 Wesbrook Mall, Vancouver, BC V6T 1Z1, Canada; 8Department of Physics, University of British Columbia, 6224 Agricultural Road, Vancouver, BC V6T 1Z1, Canada

**Keywords:** radiomics, PET/CT, saliency, head and neck cancer, outcome, HPV

## Abstract

**Simple Summary:**

This study investigated the ability of context-aware saliency-guided PET/CT radiomics in the prediction of outcome and HPV status for head and neck cancer. In total, 806 HNC patients (training vs. validation vs. external testing: 500 vs. 97 vs. 209) from 9 centers were collected from The Cancer Imaging Archive (TCIA). Saliency-guided radiomics showed enhanced performance for both outcome and HPV-status predictions relative to conventional radiomics. The radiomics-predicted HPV status also showed complementary prognostic value. This multi-center study highlights the feasibility of saliency-guided PET/CT radiomics in outcome predictions of head and neck cancer, confirming that certain regions are more relevant to tumor aggressiveness and prognosis.

**Abstract:**

Purpose: This multi-center study aims to investigate the prognostic value of context-aware saliency-guided radiomics in ^18^F-FDG PET/CT images of head and neck cancer (HNC). Methods: 806 HNC patients (training vs. validation vs. external testing: 500 vs. 97 vs. 209) from 9 centers were collected from The Cancer Imaging Archive (TCIA). There were 100/384 and 60/123 oropharyngeal carcinoma (OPC) patients with human papillomavirus (HPV) status in training and testing cohorts, respectively. Six types of images were used for radiomics feature extraction and further model construction, namely (i) the original image (Origin), (ii) a context-aware saliency map (SalMap), (iii, iv) high- or low-saliency regions in the original image (highSal or lowSal), (v) a saliency-weighted image (SalxImg), and finally, (vi) a fused PET-CT image (FusedImg). Four outcomes were evaluated, i.e., recurrence-free survival (RFS), metastasis-free survival (MFS), overall survival (OS), and disease-free survival (DFS), respectively. Multivariate Cox analysis and logistic regression were adopted to construct radiomics scores for the prediction of outcome (Rad_Ocm) and HPV-status (Rad_HPV), respectively. Besides, the prognostic value of their integration (Rad_Ocm_HPV) was also investigated. Results: In the external testing cohort, compared with the Origin model, SalMap and SalxImg achieved the highest C-indices for RFS (0.621 vs. 0.559) and MFS (0.785 vs. 0.739) predictions, respectively, while FusedImg performed the best for both OS (0.685 vs. 0.659) and DFS (0.641 vs. 0.582) predictions. In the OPC HPV testing cohort, FusedImg showed higher AUC for HPV-status prediction compared with the Origin model (0.653 vs. 0.484). In the OPC testing cohort, compared with Rad_Ocm or Rad_HPV alone, Rad_Ocm_HPV performed the best for OS and DFS predictions with C-indices of 0.702 (*p* = 0.002) and 0.684 (*p* = 0.006), respectively. Conclusion: Saliency-guided radiomics showed enhanced performance for both outcome and HPV-status predictions relative to conventional radiomics. The radiomics-predicted HPV status also showed complementary prognostic value.

## 1. Introduction

Characterizing intra-tumor heterogeneity has significant potential toward improved outcome prediction and personalized treatment planning in head and neck cancer (HNC) [1,2,3]. Accurate patient risk stratification enables personalized escalation or de-escalation of therapy, e.g., escalating radiation dose to radio-resistant areas [4] or the use of induction, concurrent, or sequential immunotherapy for high-risk patients, while implementing transoral surgery or reducing radiotherapy dose or target volumes for low-risk patients [5]. Specifically, as a distinct clinical entity of HNC, human-papillomavirus-positive (HPV+) oropharyngeal squamous cell carcinoma (OPC) patients can greatly benefit from de-escalation therapy with reduced toxicity (dysphagia, xerostomia, etc.), improving their quality of life and retaining treatment efficacy [6]. In the literature, there are several imaging techniques of HNC, but PET/CT and MRI remain the gold standard [7,8].

The field of radiomics, owing to its ability to capture sophisticated varying-order intensity-distribution patterns within tumors [9], has been extensively applied to convert medical images into minable quantitative features for different clinical tasks, including derivation of biomarkers for assessment or prediction of treatment response, survival, and therapy-related complications [10,11]. Radiomics analyses have been typically conducted on the entire tumor in original images, effectively assigning equal importance to the various pixels in computing radiomics features [12]. However, different tumor regions are biologically different in genetic and epigenetic status [13] with diverse aggressiveness and resistance, and these differences may appear as different uptake patterns (metabolism, hypoxia, proliferation) or morphological structure (neovascularization) in PET/CT images [14,15]. A number of threshold-based or clustering-based efforts [16,17] identified specific sub-regions in tumors to be promising for decision making on various cancers. Our previous study also demonstrated varying prognostic performances (C-indices: 0.52–0.69) for different sub-regions in nasopharyngeal carcinoma [18], indicating that the contribution of different regions to the outcome is different. However, these threshold- or clustering-based methods have relied on low-level vision processing, and sub-regions have been roughly discriminated by global thresholds. As such, more sophisticated or advanced computer vision processing may be valuable to determine the importance of different pixels (i.e., a weight map) prior to the computation of radiomics features.

Motivated by the field of saliency detection in computer vision [19], we hypothesize that the importance of different tumor regions can be captured by saliency calculations appropriately, as higher saliency means higher contribution (aggressiveness, uniqueness) towards the task of interest. The saliency map provides better visualization of biological distinction and may be indicative of the local risk of residual or recurrence. Thus, by involving saliency in radiomics feature extraction, model performance is likely to be improved. Moreover, the saliency map may also allow dose painting in radiotherapy [4], e.g., sub-volume boosting. There is extensive literature on saliency detection methods developed for natural image analysis [20], and these methods can be mainly divided into local, global, and local–global methods, according to the scale of features used for saliency calculation. Local-based methods can be affected by complex backgrounds, and involve too much redundant information [21], while global-based methods can suppress frequently occurring patterns and ignore useful background information [22]. Context-aware saliency detection is a local–global method [23]. It is able to not only highlight regions with distinctive patterns but also capture non-redundant background regions by using multi-scale information and achieves good performance in image retargeting and summarization.

As such, we set out to investigate the prognostic value of context-aware saliency-guided PET/CT radiomics in a large cohort multi-center setting (806 HNC patients from 9 centers). A total of six prognostic models were constructed based on (1) original PET or CT images, (2) the saliency-map itself, sub-regions with (3) higher vs. (4) lower saliency, (5) saliency-weighted images, and finally, (6) the fused PET/CT images for outcome prediction. Meanwhile, considering the fact that HPV+ OPC patients show more favorable prognoses [24] compared with HPV-OPC patients, we built six additional models and derived radiomics score for HPV-status prediction, and subsequently, the radiomics score was introduced into the aforementioned six prognostic models and the complementary prognostic value of the radiomics score of HPV-status for OPC patients was further evaluated. Besides, since model generalizability can be affected by centers, manufacturers, and voxel size [25], ComBat-based feature harmonization [26] among batches divided by these three factors was implemented, and the impact of harmonization on model performance was further evaluated. 

## 2. Materials and Methods

### 2.1. Multi-Center Data Set 

We utilized multi-center HNC data from The Cancer Imaging Archive (TCIA). Inclusion criteria included (1) histologically proven HNC, (2) the absence of distant metastases at initial diagnosis, (3) the existence of complete follow-up information, and (4) the availability of pre-treatment ^18^F-FDG PET/CT images. Exclusion criteria included (1) receiving anti-cancer therapy before PET/CT imaging or (2) having prior HNC or other kinds of cancer. This study abides by the TCIA Data Usage Policy; all datasets were de-identified, thus did not require institutional review board approval, and patient informed consent was waived.

Five HNC data collections consisting of a total of 806 patients from 9 centers were obtained: (1) The Head-Neck-PET-CT collection provided by Vallières et al., including 296 patients from 4 centers in Canada, i.e., Centre hospitalier de l’Université de Montréal (CHUM, *n* = 65), Centre hospitalier universitaire de Sherbrooke (CHUS, *n* = 100), Hôpital général juif (HGJ, *n* = 90) de Montréal, and Hôpital Maisonneuve-Rosemont (HMR, *n* = 41) de Montréal. Our previous studies [27,28] on multi-level multi-modality fusion radiomics and peri-tumor radiomics were conducted based on this collection. (2) The HNSCC collection provided by Grossberg et al. included 159 patients from the MD Anderson Cancer Center, denoted as MDACC. (3) The Head-Neck Radiomics-HN1 collection provided by Wee et al. included 74 patients from the MAASTRO Clinic in Netherlands, denoted as MAASTRO. (4) The TCGA-HNSC collection, provided by Zuley et al. from the University of Pittsburgh, included 29 patients, denoted as PITT. (5) The QIN-HEADNECK collection provided by Beichel et al. from the University of Iowa included 248 patients from two centers, denoted as QIN1 (*n* = 134) and QIN2 (*n* = 114). 

As shown in Figure 1, in our work, five centers with 597 patients (MDACC: 159, QIN1: 134, QIN2: 114, CHUS: 100, and HGJ: 90) were used for model construction (500 vs. 97 for training vs. internal validation by stratified random sampling), and the remaining 4 centers (CHUM: 65, MAASTRO: 74, HMR: 41, and PITT: 29) with 209 patients were used for external testing; the constructed radiomics score for outcome prediction (Rad_Ocm) was also separately assessed on the sub-cohort of 123 OPC patients from the external testing cohort. Furthermore, 100/384 and 60/123 OPC patients from training and external testing cohorts, respectively, whose HPV status was available, were used for HPV-status prediction model construction and testing, and the prognostic value of the radiomics score for HPV-status prediction (Rad_HPV) was evaluated on the aforementioned 123 OPC patients, comparing its usage (Rad_Ocm_HPV) vs. not using it (i.e., Rad_Ocm only).

### 2.2. PET/CT Imaging

PET/CT imaging was heterogeneous in terms of manufacturers, scanners, acquisition protocols as well as voxel sizes. For PET images, 337, 144, 182, and 143 patients were scanned by the manufacturers GE, Philips, Siemens, and CPS, respectively. Scanner types include GE Discovery (RX, ST, STE, and LS), Philips GeminiGXL 16, Siemens (Biograph TruePoint 1093, Biograph 40 TruePoint, Biograph 64 mCT, and Biograph SOMATOM Sensation-16), and CPS (1023, 1024, and Biograph HiRes Model 1080). For CT images, 206, 276, and 324 patients were scanned by GE, Philips, and Siemens, respectively. Scanner types include GE Discovery (LS, RX, ST, STE, and LightSpeed RT16), Philips (GeminiGXL 16, AcQSimCT, Brilliance 64, Mx8000 IDT 16 and PQ5000), and Siemens (Sensation 16, Emotion Duo, Biograph SOMATOM Sensation-16, Biograph 64, and Biograph 40). The in-plane resolution of PET images varied between 1.95 and 5.5 mm, while it was 0.48–1.95 mm for CT images; slice thicknesses were in the ranges of 2–5 mm and 1.5–5 mm for PET and CT images, respectively.

### 2.3. Clinical Parameters and Outcome Information 

Patients’ characteristics and follow-up information in txt or DICOM structured reporting (SR) files were collected and harmonized. Clinical information included Age, Sex, T-stage, N-stage, TNM stage, Site, Therapy, Smoking, HPV status, and Grade. The primary endpoint was disease-free survival (DFS), which was defined as the time from diagnosis to the first confirmed event of recurrence, metastasis, death, or their combination, whichever occurred first. The other three endpoints, including recurrence-free survival (RFS), metastasis-free survival (MFS), and overall survival (OS), were also evaluated. 

### 2.4. Tumor Segmentation 

For patients with available DICOM segmentations (SEG) or radiotherapy structure set (RTSTRUCT) files, the gross tumor volume of the primary tumor and lymph node (GTV and GTVnd) were extracted as CT labels and were propagated to generate PET labels. Otherwise, PET labels were first semi-automatically delineated by a radiologist (H. Z.) with 4 years of experience in head and neck nuclear medicine using the “level tracing” module in 3D Slicer 4.10.2 (https://slicer.org/, accessed on 3 November 2019), and CT labels were generated following registration. 

### 2.5. Context-Aware Saliency Map Generation

A context-aware saliency map was generated from the smallest boxes of PET and CT images that covered the entire tumor. PET images were converted to SUV. CT intensity was truncated to (–200, 300) to mitigate the interference of air and bone. As illustrated in Figure 2, the first step is to characterize the dissimilarity among patches both locally and globally. Given a patch $p_{i}$ centered at pixel i, the dissimilarity between patch $p_{i}$ and any patch $p_{j}$ can be characterized by their distance $d(p_{i},p_{j})$ as follows: (1)$$d(p_{i},p_{j})=\frac{d_{\mathrm{int}ensity}(p_{i},p_{j})}{1+c\cdot d_{position}(p_{i},p_{j})}$$
where $d_{\mathrm{int}ensity}(p_{i},p_{j})$ and $d_{position}(p_{i},p_{j})$ denote the intensity and position of Euclidean distances between patch $p_{i}$ and patch $p_{j}$, with c=3 controlling the ratio of these two distances as suggested by [23]. This dissimilarity measure is proportional to the difference in intensity and inversely proportional to the positional distance.

Since the background is likely to have similar patches at multiple scales while the dominant object is prone to having similar patches at only a few scales, in order to suppress the background, the distance calculation was set in a multi-scale manner (as also shown in Figure 2). Patch $p_{i}$ was selected from the scaled image with a scaling ratio r relative to the original image size, while patch $p_{j}$ was selected from the further scaled image with a scaling ratio s relative to the scaled image size used for patch $p_{i}$ selection. Consistent with [23], we considered 4 image scales for patch $p_{i}$: r∈{1, 0.8, 0.5, 0.3}, with corresponding patch sizes set to 7 × 7, 5 × 5, 3 × 3, and 3 × 3, and 3 image scales for patch $p_{j}$: $s\in \{r,\frac{1}{2}r,\frac{1}{4}r\}$. Thus, the multi-scale distance can be denoted as $d(p_{i}^{r},p_{j}^{s})$. 

To highlight the salient regions and suppress the non-salient ones, a salient pixel is required to be different from all the other patches, but for computational simplicity, it suffices to only consider the K most similar patches. The saliency of pixel i at scale r can be defined as: (2)$$S_{i}^{r_{}}=1-\exp\left(-\frac{1}{K}\sum_{j=1}^{K}d(p_{i}^{r},p_{j}^{s})\right)$$
where K=64, and the mean salience among different scales was calculated for each pixel i:(3)$${\bar{S}}_{i}=\frac{1}{N}\sum S_{i}^{r_{}}$$
where N=4, i.e., aforementioned 4 scales r∈{1, 0.8, 0.5, 0.3}. In order to include the immediate context, regions with mean saliency higher than 0.8 were regarded as the foci of attention, and the final saliency of each pixel was weighted by its Euclidean distance to the closest foci pixel: (4)$$S_{i}={\bar{S}}_{i}(1-d_{foci}(i))$$

As saliency maps assign a range of 0–1 weights to the pixels, we generated two sub-regions by setting a threshold of 0.5, i.e., high- and low-saliency regions, denoted as highSal and lowSal. Besides, saliency-weighted PET and CT images were also generated, and their summation was used as a fused PET-CT image to integrate important anatomical and metabolic information.

### 2.6. Feature Extraction

All images were interpolated to an isotropic voxel size of 1 × 1 × 1 mm^3^, and intensity was discretized to 64 bins before feature extraction. We utilized the Standardized Environment for Radiomics Analysis (SERA; Ver 2.1, https://github.com/ashrafinia/SERA, accessed on 30 November 2020) for feature extraction, which is in compliance with the Image Biomarkers Standardization Initiative (IBSI) [9], and further validated elsewhere [29,30]. Furthermore, 497 radiomics features were extracted from each type of image, i.e., the entire tumor region in the original PET or CT image (Origin), the saliency map itself (SalMap), the high-saliency region (highSal) or low-saliency region (lowSal) in PET or CT images according to saliency higher or lower than 0.5, and the entire tumor region in the saliency-weighted image (SalxImg) and fused image (FusedImg). The overview of the saliency-guided radiomics analysis workflow is shown in Figure 3. 

### 2.7. Reproducibility Analysis

To investigate the impact of inter- and intra-observer segmentation variations on feature reproducibility, 52 HNC patients in dataset QIN1 were segmented twice by both manual and semi-automatic methods by three experienced readers. A two-way random-effects, single-rater, absolute agreement intra-class correlation coefficient (ICC) [31] was calculated for each feature to assess inter-observer variation, and a two-way mixed-effect, single-measurement, absolute agreement ICC was calculated to evaluate intra-observer variation with the same calculation equation. Features were required to be reproducible (ICC > 0.8) for both intra- and inter-observer variations. Reproducibility was evaluated for PET and CT images, separately. For fused images, only features that were reproducible on both PET and CT images remained for subsequent analyses. 

### 2.8. Feature Harmonization

To eliminate the variation of data introduced by different acquisitions, feature harmonization was performed before model construction. In the present study, features were harmonized by ComBat [26] via different batch division strategies, assuming the batch effect is caused by three factors including the center, manufacturer, and voxel size. Thus, as shown in Appendix A, for center-based harmonization, features were harmonized among 9 centers (i.e., MDACC: 159, QIN1: 134, QIN2: 114, CHUS: 100, HGJ: 90, CHUM: 65, HMR: 41, PITT: 29, and MAASTRO: 74). For manufacturer-based harmonization, PET features were harmonized among 4 manufacturers (i.e., GE, Philips, Siemens, and CPS) and CT features were harmonized among 3 manufactures (i.e., GE, Philips, and Siemens). For voxel-size-based harmonization, PET features were grouped into 4 batches according to the in-plane resolution (88 with 1.9–3.0 mm, 383 with 3.1–3.5 mm, 174 with 3.6–4.5 mm, and 166 with 4.6–5.5 mm) since slice thickness was relatively consistent (88% within 3–4 mm), while CT features were grouped into 6 batches according to slice thickness (44 with 1.5 mm, 133 with 2.0 mm, 116 in 2.1–2.6 mm, 166 in 2.7–3.0 mm, 199 in 3.1–3.3 mm, 148 in 3.4–5.0 mm) since the in-plane resolution was relatively consistent (84% with 1 mm). Six clinical parameters (age, sex, T stage, N stage, TNM stage, and site) were introduced as biological covariates during the removal of batch effects.

### 2.9. Model Construction

In this study, as shown in Figure 3, models were constructed by considering 6 image types (Origin, SalMap, highSal, lowSal, SalxImg, or FusedImg), 3 involved modalities (Clinical and/or PET and/or CT), and 4 harmonization methods (none, center-, manufacturer-, or voxel size-based). 

In each model configuration, univariate Cox analysis was first used to evaluate the prognostic performance of each feature in the training cohort, i.e., calculating 4 C-indices for each feature when predicting 4 endpoints (RFS, MFS, OS, and DFS). Thus, the mean C-index was used to characterize the overall performance of each feature, and the top 20 non-redundant features with the highest mean C-index values were selected as candidate features for subsequent multivariate Cox model construction. Starting with the first feature, the multivariate Cox model was built via stepwise addition of the remaining features, i.e., 19 models were constructed, and it was only conducted for primary endpoint DFS prediction (i.e., only one C-index per model) in the training cohort instead of for each endpoint to avoid overfitting. Then, for each constructed model, 3 C-indices for 3 secondary endpoints (RFS, MFS, and OS) predictions in the training cohort and 4 C-indices for all endpoints predictions in the validation cohort were calculated. As such, the model showing the highest mean C-index (averaging among these 8 C-indices) in both training and validation cohorts for all four endpoint prediction tasks was selected. Features retained in this model weighted by their corresponding coefficients were regarded as the radiomics score for outcome prediction (Rad_Ocm). Therefore, only six selected models (one model per image type) were tested in the independent external testing set, consistent with AI algorithm development best-practice guidelines that models should be tested on an unseen holdout dataset [32]. 

Furthermore, considering that HPV status was only available for 100 OPC patients in the training cohort, by using the same configurations of these selected 6 prognostic models, 6 multivariate logistic regression models were constructed for HPV status prediction. Similar to the outcome prediction model construction, features were first sorted by their AUC values in descending order, the top 2–20 non-redundant features were then fed into multivariate logistic models, and finally, the model with higher AUC and lower Bayesian information criterion (BIC) values was selected (by comparing the summation of ranks of descending AUC and ascending BIC). Thus, the radiomics score for HPV status prediction (Rad_HPV) can be obtained in the same way. Next, 60 OPC patients from the external testing cohort whose HPV status was available were used to test the performance of the HPV-status prediction model. Since HPV status is well-known to be relevant in the prognosis of OPC, the complementary prognostic value of Rad_HPV was further evaluated in 123 OPC patients from the testing cohort, comparing its usage (Rad_Ocm_HPV) vs. not using it (i.e., Rad_Ocm only) (Figure 1).

The model performance was reported by the C-index and AUC with a 95% confidence interval using 100 bootstraps, and was further evaluated regarding Kaplan–Meier curves, the log-rank test (the median value of radiomics score from the training cohort was used for separating high- or low-risk group), and the receiver operating characteristic curve (ROC) when appropriate.

## 3. Results

### 3.1. Patient Characteristics

In total, 806 patients from 9 different centers were enrolled (Table 1), including 654 (81%) men and 161 (19%) women, with a median age of 60 (18–91) years old. There were 28, 60, 148, and 570 patients with overall clinical stage I, II, III, and IV, respectively. The primary tumor sites were the nasopharynx, oropharynx, hypopharynx, larynx, oral cavity, and other, for 37, 507, 29, 144, 57, and 32 patients, respectively. Patients received surgery, radiotherapy (RT), or concurrent chemo-radiotherapy (CRT) following initial staging. Among these 806 patients, only 201 (25%) patients had HPV status (*n* = 115 and 86 for positive and negative), and 160 patients of them had OPC. Furthermore, 368 (46%) patients had graded tumors (*n* = 17, 212 and 139 for grade 1, 2, and 3). The median follow-up time was 50 months (range 1–133 months), and 215 (27%), 168 (21%), and 244 (30%) patients experienced tumor recurrence, metastasis, and death, respectively. 

### 3.2. Feature Screening

The mean (range) inter- and intra-observer ICCs for the 497 PET features were 0.83 (0.17–0.99) and 0.84 (0.35–0.99), respectively, while it was 0.85 (0.04–0.99) and 0.86 (0.19–0.99) for 497 CT features. There were 322 PET features and 406 CT features with both inter- and intra-observer ICCs > 0.8 that remained reproducible with segmentation (Appendix A). Following redundancy removal there were 54 PET features and 62 CT features that remained for subsequent model construction. The top 20 non-redundant PET and CT features showed C-indices of 0.53–0.60 and 0.53–0.63 for DFS prediction, respectively, in 500 training patients. There are 28/62 non-redundant features that showed AUC > 0.6 for HPV status prediction in 100 OPC HPV training patients. 

### 3.3. Performance of Saliency Guided Prognostic Model

The C-indices of the six selected optimal models in external testing cohorts are listed in Table 2, and the corresponding model configurations (image type, modality, and harmonization method) and the number of features are also provided. Compared with the Origin model, SalMap and SalxImg models achieved the highest C-indices for RFS (0.621 vs. 0.559) and MFS (0.785 vs. 0.739) predictions, respectively; the FusedImg model performed the best for both OS (0.685 vs. 0.659) and DFS (0.641 vs. 0.582) predictions. The performance of these six selected models in training and validation cohorts is provided in Appendix A.

Among the six selected optimal models listed in Table 2, none of them were constructed by the center-based harmonization method, while four models achieved their best performance under manufacturer- or voxel size-based harmonization methods, indicating the batch effect may not be prominently expressed over different centers but different manufacturers and voxel sizes, thus suppressing the effect of center-based harmonization on performance improvement. 

Figure 4 shows the K-M curves of the Origin model and selected representative saliency-guided models in the external testing cohort. All saliency-related models showed higher hazard ratio (HR) values than the Origin model for the four outcome predictions (1.60–6.77 vs. 1.25–4.36). High-risk patients identified by the SalxImg model showed a 5-year MFS value of 61% (Figure 4f), while this value was 67% in the Origin model (Figure 4b). The FusedImg model achieved significant separation between high- and low-risk groups for DFS prediction with a *p*-value of 0.003 (Figure 4h), while it was 0.048 for the Origin model (Figure 4d).

### 3.4. Complementary Prognostic Value of Radiomics Score of HPV Status

Under the same model configuration, the radiomics score for HPV-status prediction (Rad_HPV) was built based on 100 OPC patients from the training cohort. As shown in Figure 5, Rad_HPV of the Origin model and the FusedImg model showed similar AUC values of 0.706 and 0.702 for HPV-status prediction in the training set, while the FusedImg model showed a higher AUC value of 0.653 compared with the Origin model with an AUC value of 0.484 in the testing set. 

The prognostic value of Rad_HPV was further evaluated on 123 OPC patients from the external testing cohort (Table 3 and Appendix A). Compared with the radiomics score for outcome prediction alone (Rad_Ocm) or their combination (Rad_Ocm_HPV), Rad_HPV values of FusedImg and SalxImg models showed the highest C-indices of 0.641 and 0.687 for RFS and MFS predictions, respectively, while Rad_Ocm_HPV of the FusedImg model performed the best for OS and DFS predictions with C-indices of 0.702 and 0.684 and log-rank *p*-values of 0.002 and 0.006, respectively (Figure 6). The features retained in the SalMap, SalxImg, and FusedImg models and the corresponding coefficients are reported in Appendix A.

### 3.5. Saliency Distribution Localization 

For a better understanding of the contribution of regions with different saliency values, we calculated the normalized distance between the highest saliency voxel and the centroid in PET and CT images. A distance of 0 indicates the highest saliency located at the centroid, while 1 represents its location at the farthest edge. As shown in Figure 7, the highest saliency of more than half of the patients was located far from the centroid with distances ranging from 0.6–0.9, while a small portion of patients showed higher saliency close to the centroid. 

## 4. Discussion

This study investigated the prognostic value of saliency-guided PET/CT radiomics (SalMap, highSal, lowSal, SalxImg, and FusedImg) in HNC, and models were developed and validated on 806 patients from 9 centers. Our results demonstrate the great prognostic potential of saliency radiomics (SalMap, SalxImg, and FusedImg) compared with the conventional radiomics model (Origin), especially for MFS, OS, and DFS prediction. This is analogous with previous study findings that showed fused PET/CT radiomics outperformed PET or CT alone [27,33], e.g., when adopting wavelet-based fusion. Besides, saliency radiomics also outperformed conventional radiomics for HPV status prediction, and integrating Rad_HPV with Rad_Ocm showed a complementary prognostic value compared with only involving Rad_Ocm in a sub-cohort of OPC patients. 

Vuong et al. [34] created a radiomics feature activation map in CT images of non-small-cell lung cancer by comparing whether the feature value from a small patch was higher than the population median feature value from the entire tumor or not, aiming to track the spatial location of regions responsible for signature activation. They found that the texture feature GLSZM_zone size non-uniformity normalized was more activated on the adjacent region of the tumor. By contrast, the saliency map in our present study does not need population information and can be generated for any new patient due to the internal comparison mechanism. The saliency map provides the visual perception of the prognostic contribution of each region, and high saliency was found to be more frequently located at the remoter region relative to the centroid for both PET and CT images in our study, indicating the aggressiveness and special biological behavior of frontier area. Nevertheless, more intuitive comparisons of different regions may be obtained when both pre- and post-therapy or follow-up images are available, and thus shrinkage and resistance or recurrence regions can be identified [35,36]. 

HPV status has been documented as an independent prognostic factor in OPC [24], and several studies have demonstrated the predictive ability of PET/CT radiomics features in HPV-status prediction [37,38,39]. This has motivated us to speculate that radiomics-predicted HPV status might also be prognostic for outcomes. In this study, given the limited number of patients who are available regarding HPV status, it was not feasible to construct prognostic models for the entirety of the model by incorporating HPV status as a predictor directly, while abandoning the HPV status arbitrarily means sacrificing important information. As such, prognostic information of HPV was carried by a radiomics score that was predictive of HPV status, and Rad_HPV was further combined with Rad_Ocm for outcome prediction. Better prognostic performance was achieved in the OPC testing cohort, confirming the effectiveness of this strategy. 

In a multi-center study, an unavoidable problem is the so-called batch effect where radiomics features are sensitive to the manufacturer, acquisition protocol, and reconstruction algorithm. A compensation method was originally proposed in genomics [40] and extensively applied in radiomics based on different batch divisions [41,42]. Orlhac et al. [41] found that ComBat can remove the batch effect caused by different reconstruction algorithms, kernels, and slice thicknesses in CT images with lung cancer, and also found ComBat to enhance the identification performance of triple-negative breast cancer in PET images obtained from two different centers [42]. In the present study, different manufacturers and voxel sizes were observed in the same center (Appendix A). As such, different batch divisions can be generated based on centers, manufacturers, and voxel size to fully explore the effect of feature harmonization (ComBat) on model generalization. Our results (Table 2) showed that different batch divisions result in varying effects on different outcome predictions and model configurations; in fact, for Origin and SalMap models, an even lower performance was obtained after harmonization compared with no harmonization. One possible reason is that ComBat assumes additive and multiplicative batch effects satisfying Gaussian and inverse Gamma distributions [40], while, due to the existence of complex confounders, this strict assumption does not hold in some circumstances, especially for a variety of radiomics features. More effective batch-division strategies are warranted to obtain the prominent factors of variability and ensure the prerequisite of ComBat. Other advanced harmonization strategies [43] may allow for better generalization, e.g., Modified ComBat (M-ComBat) [44] by harmonizing features to the mean and standard deviation of a selected reference batch instead of to the global mean and standard deviation of all batches, Bootstrapped ComBat (B-ComBat) by estimating parameters in a bootstrapped manner to improve the robustness, or their combination (BM-ComBat). Deep-learning-based harmonization [45] in the image domain may also provide another solution. 

The present study has some limitations. First, the dataset was retrospectively collected from the TCIA public platform, wherein each dataset may be originally collected for different purposes, and patient inclusion bias may have existed; thus, prospective cohorts can be studied for further validation in the future. Second, we used the default parameter setting instead of extensive tuning when constructing the saliency map to avoid overfitting, though parameter tuning may result in better performance. Third, the saliency map was generated slice by slice, while 3D methods may produce more precise saliency information and still need to be investigated. Fourth, investigating other saliency detection methods or designing a dedicated saliency detection method that is more appropriate to characterize the importance of the tumor sub-region in PET/CT HNC is of interest in the future. Finally, HPV-status prediction was only conducted on OPC patients, and the prognostic value of predicted HPV status on tumors located at other sites (e.g., larynx and hypopharynx) remains to be explored. 

## 5. Conclusions

Our multi-center study highlights the feasibility of saliency-guided PET/CT radiomics in outcome prediction of head and neck cancer, confirming that certain regions are more relevant to tumor aggressiveness and prognosis. The radiomics score as a surrogate of HPV status also conveyed prognostic information. Feature harmonization among batches divided by the manufacturer and voxel size showed better performance improvement compared with that divided by centers.

## Figures and Tables

**Figure 1 cancers-14-01674-f001:**
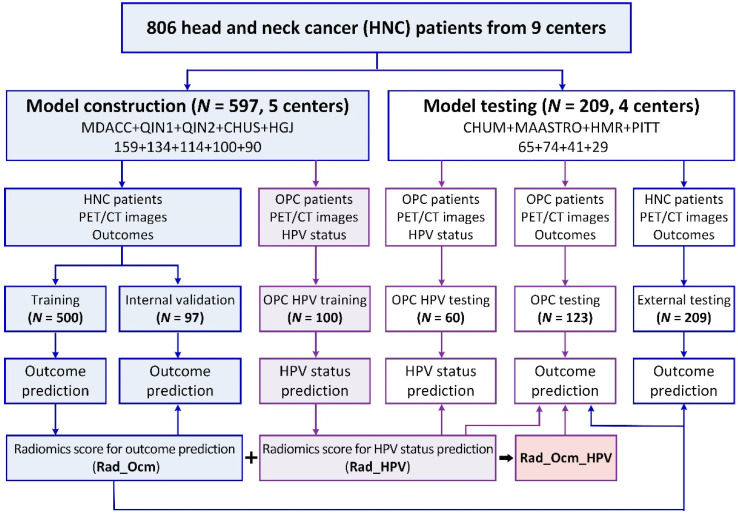
Our model construction and testing paradigms in this radiomics study.

**Figure 2 cancers-14-01674-f002:**
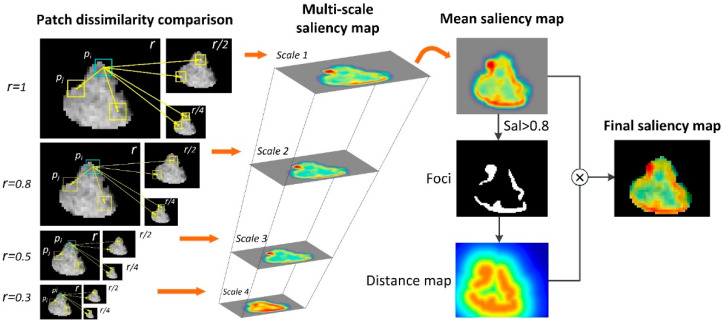
Schematic illustration of context-aware saliency map generation. Patch dissimilarity was compared in a multi-scale manner to generate a saliency map at each scale. The final saliency map was then constructed by multiplying mean saliency map and distance map.

**Figure 3 cancers-14-01674-f003:**
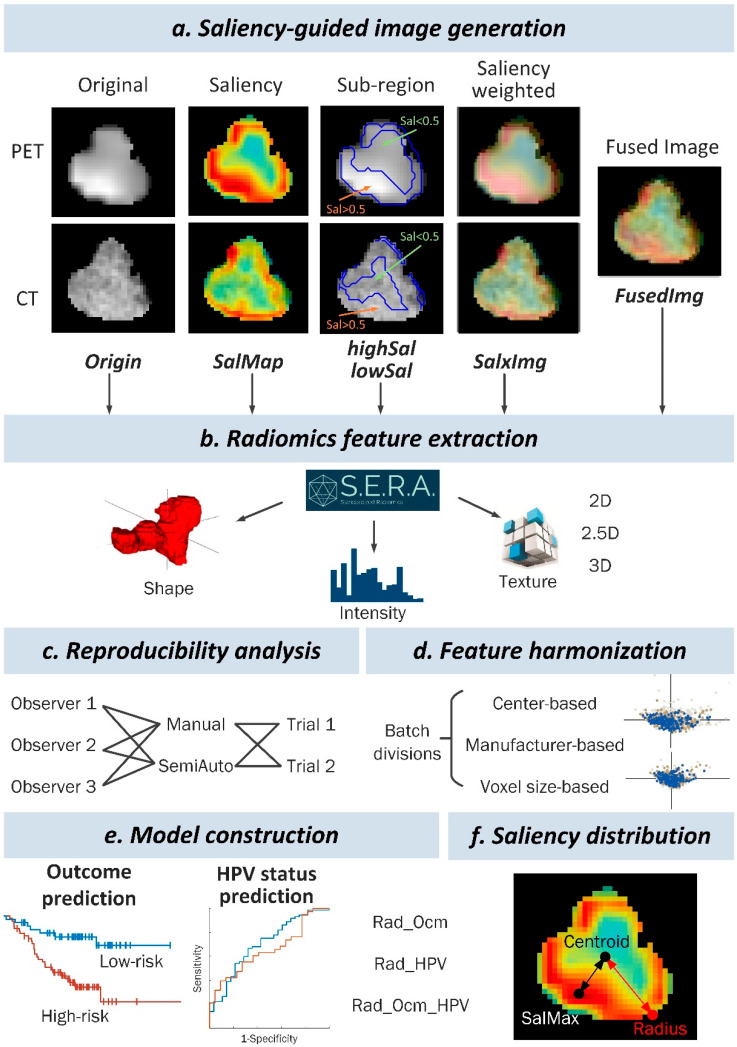
Overview of the investigated saliency-guided radiomics analysis workflow, including saliency-guided image generation (Origin, SalMap, highSal, lowSal, SalxImg, and FusedImg), radiomics feature extraction, reproducibility analysis, feature harmonization, model construction (outcome prediction and HPV status prediction), and saliency distribution.

**Figure 4 cancers-14-01674-f004:**
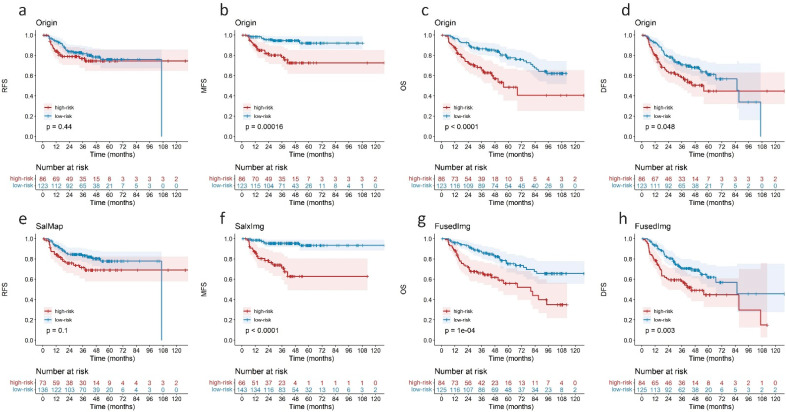
The K-M curves of (**a**–**d**) Origin model and (**e**–**h**) selected representative saliency-guided models from Table 2 in external testing cohort (209 patients).

**Figure 5 cancers-14-01674-f005:**
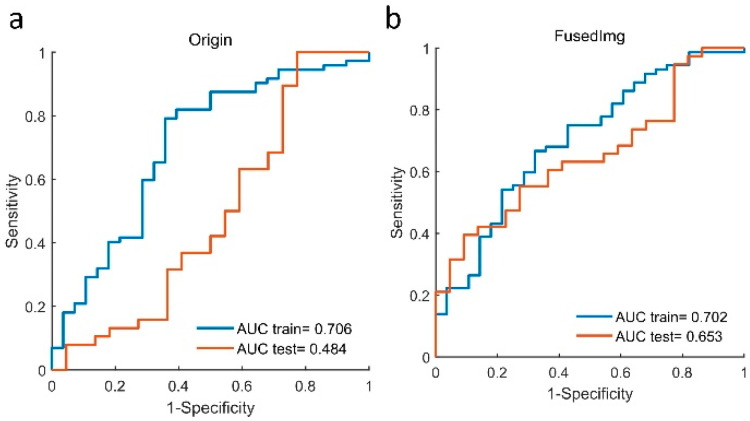
AUC value of HPV status prediction for (**a**) Origin model and (**b**) FusedImg model in both training and testing cohorts with 100 and 60 patients, respectively.

**Figure 6 cancers-14-01674-f006:**
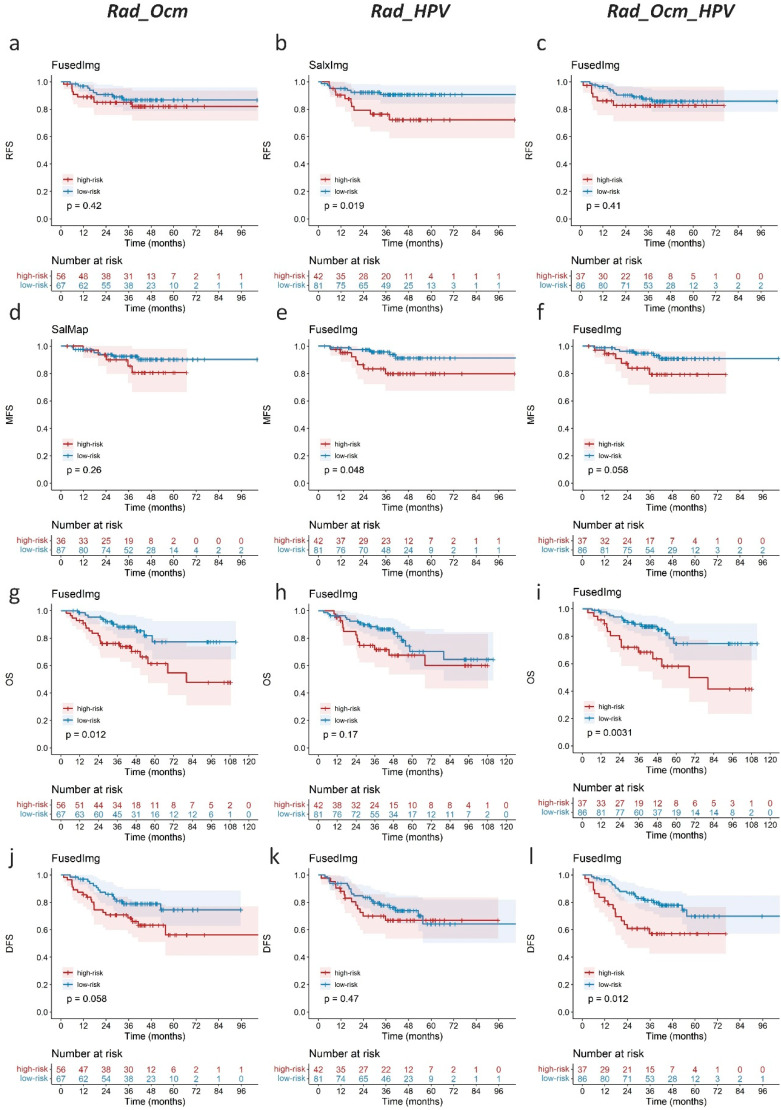
The K-M curves of (**a**,**d**,**g**,**j**) Rad_Ocm, (**b**,**e**,**h**,**k**) Rad_HPV, and (**c**,**f**,**i**,**l**) Rad_Ocm_HPV model for RFS, MFS, OS, and DFS predictions in 123 OPC patients from testing cohort as listed in Table 3.

**Figure 7 cancers-14-01674-f007:**
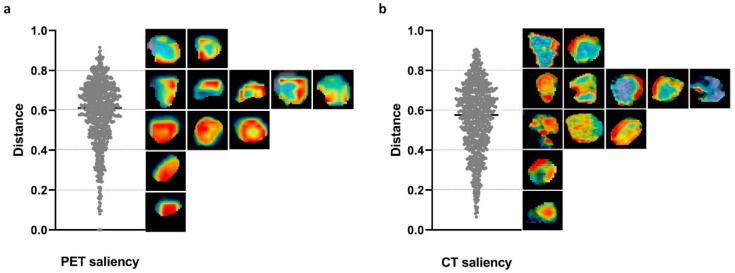
The distribution of distance between the voxel with highest saliency and the centroid in (**a**) PET and (**b**) CT images.

**Table 1 cancers-14-01674-t001:** Clinical characteristics of the HNC patients in training (*n* = 500), internal validation (*n* = 97), and external testing (*n* = 209) cohorts. There are 123 patients from external testing cohort with OPC (OPC testing), and amongst them, 60 patients had HPV status (OPC HPV testing). Furthermore, 100 OPC patients from the training cohort had HPV status (OPC HPV training).

Characteristic	All	Training	Internal Validation	External Testing	OPC HPV Training	OPC HPV Testing	OPC Testing
No.	806	500	97	209	100	60	123
Age, median (range)	60 (18–91)	58 (18–90)	57 (20–82)	63 (26–90)	62 (34–81)	60 (44–76)	63 (38–90)
Sex, no. (%)							
Male	654 (81%)	404 (81%)	78 (80%)	163 (78%)	82 (82%)	47 (78%)	93 (76%)
Female	161 (19%)	96 (19%)	19 (20%)	46 (22%)	18 (18%)	13 (22%)	30 (24%)
Smoking, no. (%)							
Non-smoker	122 (15%)	87 (17%)	25 (26%)	10 (5%)	16 (16%)	2 (3%)	3 (2%)
Former-smoker	107 (13%)	70 (14%)	27 (28%)	10 (5%)	3 (%)	2 (3%)	5 (4%)
Current-smoker	207 (26%)	153 (30%)	45 (46%)	9 (4%)	10 (%)	-	3 (2%)
TNM Stage, no. (%)							
I	28 (3%)	7 (2%)	3 (3%)	18 (9%)	2 (%)	3 (5%)	4 (3%)
II	60 (8%)	36 (7%)	8 (8%)	16 (8%)	8 (%)	3 (5%)	7 (6%)
III	148 (18%)	106 (21%)	13 (14%)	29 (13%)	14 (%)	7 (12%)	14 (11%)
IV	570 (71%)	351 (70%)	73 (75%)	146 (70%)	76 (%)	47 (78%)	98 (80%)
Site, no. (%)							
Nasopharynx	37 (5%)	29 (6%)	-	8 (4%)	-	-	-
Oropharynx	507 (63%)	317 (63%)	67 (69%)	123 (60%)	100 (100%)	60 (100%)	123 (100%)
Hypopharynx	29 (3%)	18 (4%)	2 (2%)	9 (4%)	-	-	-
Larynx	144 (18%)	85 (17%)	6 (7%)	53 (25%)	-	-	-
Oral cavity	57 (7%)	29 (6%)	17 (18%)	11 (5%)	-	-	-
Others	32 (4%)	22 (4%)	5 (5%)	5 (2%)	-	-	-
Therapy, no. (%)							
Surgery	35 (4%)	1 (1%)	9 (9%)	25 (12%)	-	4 (7%)	10 (8%)
RT	141 (18%)	69 (13%)	3 (3%)	69 (33%)	16 (%)	20 (33%)	29 (24%)
Surgery + RT	42 (5%)	30 (6%)	7 (7%)	5 (2%)	-	-	-
CRT	508 (63%)	334 (67%)	67 (69%)	107 (51%)	77 (%)	35 (58%)	82 (67%)
Surgery + CRT	80 (10%)	66 (13%)	11 (12%)	3 (2%)	7 (%)	1 (2%)	2 (1%)
HPV status, no. (%)							
Positive	115 (14%)	76 (15%)	-	39 (18%)	72 (%)	38 (63%)	38 (31%)
Negative	86 (11%)	55 (11%)	-	31 (14%)	28 (%)	22 (37%)	22 (18%)
Grade, no. (%)							
1	17 (2%)	15 (3%)	2 (2%)	-	2 (%)	-	-
2	212 (26%)	148 (30%)	47 (48%)	17 (8%)	8 (%)	2 (3%)	8 (7%)
3	139 (17%)	104 (21%)	24 (25%)	11 (5%)	17 (%)	2 (3%)	3 (2%)
Outcome, no. (%)							
Recurrence	215 (27%)	134 (27%)	36 (37%)	45 (22%)	12 (%)	7 (12%)	18 (15%)
Metastasis	168 (21%)	115 (23%)	27 (28%)	26 (12%)	13 (%)	4 (7%)	12 (10%)
Death	244 (30%)	151 (30%)	27 (28%)	66 (32%)	10 (%)	16 (27%)	29 (24%)

OPC: Oropharyngeal carcinoma; HPV: Human papillomavirus; RT: radiotherapy; CRT: Chemoradiotherapy.

**Table 2 cancers-14-01674-t002:** C-index (95% confidence interval), model configuration (including image type, modality, and harmonization method), and number of features for each model in the external testing cohorts. Best performances are also highlighted.

Model Configuration				External Testing C-Index			
Image Type	Modality	Harmonization Method	Num of Features	RFS	MFS	OS	DFS
Origin	CT	None	3	0.559	0.739	0.659	0.582
				(0.459–0.654)	(0.637–0.843)	(0.587–0.739)	(0.508–0.661)
SalMap	Clin + CT	None	8	0.621	0.759	0.677	0.636
				(0.518–0.721)	(0.668–0.848)	(0.604–0.756)	(0.562–0.713)
highSal	CT	Scanner-based	20	0.575	0.676	0.600	0.563
				(0.469–0.679)	(0.572–0.787)	(0.524–0.681)	(0.483–0.645)
lowSal	CT	Scanner-based	4	0.604	0.778	0.663	0.612
				(0.499–0.705)	(0.689–0.866)	(0.599–0.732)	(0.539–0.687)
SalxImg	PET + CT	Voxel size-based	2	0.616	0.785	0.675	0.633
				(0.537–0.688)	(0.712–0.864)	(0.614–0.740)	(0.579–0.688)
FusedImg	PET + CT	Voxel size-based	3	0.595	0.746	0.685	0.641
				(0.506–0.690)	(0.669–0.837)	(0.626–0.761)	(0.575–0.718)

**Table 3 cancers-14-01674-t003:** The highest C-index (95% confidence interval) of Rad_Ocm, Rad_HPV, and Rad_Ocm_HPV for outcome prediction in 123 OPC patients from testing cohort; models were selected from Appendix A.

	Rad_Ocm		Rad_HPV		Rad_Ocm_HPV	
	C-Index	Model	C-Index	Model	C-Index	Model
RFS	0.603	FusedImg	0.641	SalxImg	0.616	FusedImg
	(0.460–0.737)		(0.544–0.749)		(0.491–0.731)	
MFS	0.668	SalMap	0.687	FusedImg	0.683	FusedImg
	(0.505–0.822)		(0.546–0.853)		(0.544–0.843)	
OS	0.671	FusedImg	0.627	FusedImg	0.702	FusedImg
	(0.553–0.781)		(0.529–0.738)		(0.599–0.807)	
DFS	0.651	FusedImg	0.615	FusedImg	0.684	FusedImg
	(0.546–0.752)		(0.533–0.717)		(0.589–0.780)	

## Data Availability

The data presented in this study are available on request from the corresponding author.

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
