# Peer review of "Context-Aware Saliency Guided Radiomics: Application to Prediction of Outcome and HPV-Status from Multi-Center PET/CT Images of Head and Neck Cancer"

_cancers, 2022, doi:10.3390/cancers14071674_

Round 1
Reviewer 1 Report
This study looks at how saliency guided PET/CT radiomics can improve the prediction of outcome and HPC status for HNC cancer patients.
The greatest strength of the paper, in my opinion, is the extremely thorough and painstaking attention to detail. The assessments of nonredundancy, robustness, etc... are very much appreciated. I think if more papers in the radiomics literature did this, the field would be much better off. The one scientific addition I do request from the authors is to include more details on the uncertainties and confidence intervals of the numerous numbers they report. I need this information to get a better sense of how significant the conclusions are.
That said, while I personally love the rigor and attention to detail, in my opinion it comes at the expense of obscuring the novelty and significance of the results. After reading the paper multiple times, I still don't have a good sense of what saliency guided radiomics means, or what the essential punch line of the results is. I would ask the authors to add substantially more `meat' to the introduction and discussion sections, similar to what they have in the methods and results sections currently. If space is an issue, perhaps some of the details in the methods and results could be moved to the supplement.
Author Response
We truly appreciate the reviewer’s time and positive comments about the innovation, comprehensibility and structure of our work. Below we address points one by one, and also highlight the changes in the manuscript itself.
Reviewer 1
This study looks at how saliency guided PET/CT radiomics can improve the prediction of outcome and HPV status for HNC cancer patients.
The greatest strength of the paper, in my opinion, is the extremely thorough and painstaking attention to detail. The assessments of nonredundancy, robustness, etc... are very much appreciated. I think if more papers in the radiomics literature did this, the field would be much better off. The one scientific addition I do request from the authors is to include more details on the uncertainties and confidence intervals of the numerous numbers they report. I need this information to get a better sense of how significant the conclusions are.
Thank you for your great suggestions. We have now provided the 95% confidence interval of C-index and AUC values in all Tables, Figure 4 and 6 were also updated, and this sentence “Model performance was reported by C-index and AUC with 95% confidence interval using 100 bootstrap” was also added in Methods and Methods.
That said, while I personally love the rigor and attention to detail, in my opinion it comes at the expense of obscuring the novelty and significance of the results. After reading the paper multiple times, I still don't have a good sense of what saliency guided radiomics means, or what the essential punch line of the results is. I would ask the authors to add substantially more ‘meat’ to the introduction and discussion sections, similar to what they have in the methods and results sections currently. If space is an issue, perhaps some of the details in the methods and results could be moved to the supplement.
Thank you for your comments. We have now added more explanations in Introduction: “Since different tumor regions are biologically different in genetic and epigenetic status [13] with diverse aggressiveness and resistance, and these differences may appear as different uptake patterns (metabolism, hypoxia, proliferation) or morphological structure (neovascularization) in PET/CT images [14, 15]. Motivated by the field of saliency detection in computer vision [19], we hypothesize that the importance of different tumor regions can be captured by saliency calculations appropriately, as higher saliency means higher contribution (aggressiveness, uniqueness) towards the task of interest. The saliency map provides better visualization of biological distinction and may be indicative of the local risk of residual or recurrence. Thus, by involving saliency into radiomics feature extraction, the model performance is likely to be improved. Beside, the saliency map may also allow dose painting in radiotherapy [4], e.g. sub-volume boosting.” and Discussion: “Forth, investigating some other saliency detection methods or designing dedicated saliency detection method that is more appropriate to character the importance of tumor sub-region in PET/CT HNC is of research interest in the future.”
In our experiment, a total of 6 prognostic models were constructed based on original (1) PET or CT images, (2) saliency-map itself, sub-regions with (3) higher vs. (4) lower saliency, (5) saliency weighted images and finally (6) the fused PET/CT images for outcome prediction. Thus “saliency guided” has multiple meanings, first, it means that we can extract radiomics features from the saliency map itself instead of from original images, besides, two sub-regions with higher or lower saliency value were also determined by saliency map, and radiomics features were extracted from the sub-regions instead of the whole region of the tumor. Third, saliency map was regarded as a weight map, the saliency value means the importance of each pixel, and radiomics features were extracted from the saliency weighted images. Finally, two saliency weighted images (PET and CT) were further fused, and radiomics features were also extracted from the fused image. We used the word “saliency guided” because saliency was involved in the feature extraction.

Reviewer 2 Report
The Authors investigated the role of saliency map in order to provide visualization of biological characteristics of tumor subegions. The authors performed a multicentriuc study to explore the feasibility of saliency-guided PET/CT radiomics for outcome prediction in head and neck cancer patients. The study is rather interesting. Methodology robust and scientifically sound. The authors should be acknowledged for their effort. Minor comments:
Abstract
- Line 34-36. Sounds awkward. Please rephrase.
- Line 39. Please state the textual outcome and then use the acronym between brackets (OS) (DFS).
Author Response
Reviewer 2
The Authors investigated the role of saliency map in order to provide visualization of biological characteristics of tumor subegions. The authors performed a multicentriuc study to explore the feasibility of saliency-guided PET/CT radiomics for outcome prediction in head and neck cancer patients. The study is rather interesting. Methodology robust and scientifically sound. The authors should be acknowledged for their effort. Minor comments:
Thank you for your interest in our work and positive comments. Below we address points one by one.
Abstract
Line 34-36. Sounds awkward. Please rephrase.
Thank you for your suggestion. It was now rephrased as “Six types of images were used for radiomics feature extraction and further model construction, namely (i) original image (Origin), (ii) context-aware saliency map (SalMap), (iii, iv) high or low saliency regions in original image (highSal or lowSal), (v) saliency weighted image (SalxImg), and finally (vi) the fused PET-CT image (FusedImg).”
Line 39. Please state the textual outcome and then use the acronym between brackets (OS) (DFS).
Thank you for your suggestion. It was now stated as “Four outcomes were evaluated, i.e. recurrence-free survival (RFS), metastasis-free survival (MFS), overall survival (OS) and disease-free survival (DFS), respectively.”

Reviewer 3 Report
Dear authors,
The article you submitted for publication entitled: "Context-Aware Saliency Guided Radiomics: Application to Prediction of Outcome and HPV-Status from Multi-Center PET / CT 3 Images of Head and Neck Cancer "is fascinating, but above all, ambitious. I have read the text thoroughly, and I believe there is a need to carry out some "minor revisions" before publication.
1. Line 112: "was further eval." Please complete the phrase with the missing word.
2. When discussing oral imaging techniques, it should be noted that there are several techniques today. Try adding this sentence in the introduction (after treatment efficacy [6]): "In the literature there are several imaging techniques of HNC, but PET / CT and MR remain the gold standard ( 1. Cicciù M et al. Early Diagnosis on Oral and Potentially Oral Malignant Lesions: A Systematic Review on the VELscope® Fluorescence Method. Dent J (Basel). 2019; 7 (3): 93. Doi: 10.3390 / dj7030093 / 2. Di Stasio D, et al. High-Definition Ultrasound Characterization of Squamous Carcinoma of the Tongue: A Descriptive Observational Study. Cancers (Basel). 2022 Jan 23; 14 (3): 564. doi: 10.3390 / cancers14030564. PMID: 35158831; PMCID: PMC8833637. )
3. English needs to be revised
Author Response
Reviewer 3
Dear authors,
The article you submitted for publication entitled: "Context-Aware Saliency Guided Radiomics: Application to Prediction of Outcome and HPV-Status from Multi-Center PET / CT 3 Images of Head and Neck Cancer "is fascinating, but above all, ambitious. I have read the text thoroughly, and I believe there is a need to carry out some "minor revisions" before publication.
We wish to thank the reviewer’s time and positive comments. Below we address points one by one.
- Line 112: "was further eval." Please complete the phrase with the missing word.
Thank you for noticing this. We have rephrased it as “was further evaluated”
- When discussing oral imaging techniques, it should be noted that there are several techniques today. Try adding this sentence in the introduction (after treatment efficacy [6]): "In the literature there are several imaging techniques of HNC, but PET / CT and MR remain the gold standard ( 1. Cicciù Met al. Early Diagnosis on Oral and Potentially Oral Malignant Lesions: A Systematic Review on the VELscope® Fluorescence Method. Dent J (Basel). 2019; 7 (3): 93. Doi: 10.3390 / dj7030093 /2. Di Stasio D, et al. High-Definition Ultrasound Characterization of Squamous Carcinoma of the Tongue: A Descriptive Observational Study. Cancers (Basel). 2022 Jan 23; 14 (3): 564. doi: 10.3390 / cancers14030564. PMID: 35158831; PMCID: PMC8833637. )
Thank you for your suggestions. We have now added this sentence in introduction and cited the two references.
- English needs to be revised
Thank you for your suggestion. We have checked the grammar and syntactical errors throughout the manuscript for better understanding.
